# Zinc Oxide Nanoparticles Enhanced Growth of Tea Trees via Modulating Antioxidant Activity and Secondary Metabolites

**Chen Chen** [1,2,†], **Jiaying Lai** [3,†], **Hong Chen** [2] and **Fangyuan Yu** [2,*]

1 School of Landscape and Horticulture, Yangzhou Polytechnic College, Yangzhou 225009, China; cc0212@njfu.edu.cn
2 College of Forest Science, Nanjing Forestry University, Nanjing 210037, China; hongchen@njfu.edu.cn
3 Key Laboratory of Plant Resources Conservation and Sustainable Utilization, South China Botanical Garden, Chinese Academy of Sciences, Guangzhou 510650, China; sixjie@njfu.edu.cn
* Correspondence: lullabies@njfu.edu.cn
† These authors contributed equally to this work.

**Abstract:** Nano-fertilizer has been dubbed 'the fertilizer of the 21st century', and it is already being used extensively in agriculture. Zinc oxide nanoparticles (ZnO-NPs) have excellent biological properties and are expected to be an ideal choice for plant zinc fertilizer. Tea is one of the top three beverages in the world, and improving the quality of tea is a priority in its research field. In this study, different concentrations (0, 10, 50, 100, 150, and 200 mg·L$^{-1}$) of ZnO-NPs were sprayed on tea leaves to investigate their effects on volatile aroma substances and biochemical aspects of tea leaves. The results revealed that various concentrations of ZnO-NPs had different effects on physiological indexes. The concentration of 150 mg·L$^{-1}$ of ZnO-NPs enhanced chlorophyll content, while the 100 mg·L$^{-1}$ concentration of ZnO-NPs promoted the accumulation of soluble proteins and the activity of antioxidant enzymes, including a decrease in the content of malondialdehyde. In addition, the ZnO-NPs spray reduced the content of tea polyphenols. A total of 27 volatiles were identified under six treatments, with benzene being the common compound with an average content of 45.97%. Ethanolamine and cis-3-hexenyl acetate were the other two major compounds. It was concluded that the presence of ZnO-NPs improved the antioxidant system of teas, increased soluble protein content and provided better reactive oxygen species protection for plants, especially in the case of ZnO-NPs at 100 mg·L$^{-1}$. We highlighted that ZnO-NPs application was a favorable way to improve tea trees growth.

**Keywords:** antioxidant enzymes; *Camellia sinensis*; physiological parameters; volatile compounds; zinc oxide nanoparticles





## 1. Introduction

Nanotechnology is the branch of science that studies different types of nanoparticles (NPs), ranging in size from 1 to 100 nm [1]. It is also the most progressing field of science, providing researchers with new tools to apply NPs to plants. Application of nanotechnology include improvement of agricultural production with bioconjugated NPs, the cultivation of pest-resistant plant varieties, and the use of foliar fertilizer to improve crop growth [2]. In recent years, NPs have been extensively applied in commercial applications and are gradually being developed into the agriculture [3]. Different types of NPs such as zinc oxide nanoparticles (ZnO-NPs), gold nanoparticles (Au-NPs), copper nanoparticles (CuO-NPs), silver nanoparticles (AgNO$_3$-NPs), and titanium dioxide nanoparticles (TiO$_2$-NPs) have been used by researchers for the growth and development of plants [4].

Among these NPs, ZnO-NPs have received much more attention. It is estimated that 33,400 tons of ZnO-NPs are produced globally, making them the third most universally used NPs [5]. The levels of ZnO-NPs in environment range from 3.1 to 31 μg·kg$^{-1}$ in soil and 76–760 μg·kg$^{-1}$ in water [6]. ZnO-NPs are considered to be a bio-safe material with photo-oxidative and photo-catalytic activities towards biological species [7]. In addition, ZnO-NPs

also contributed to plant growth and development. The growth performance and dry weight of *Cucumis sativus* increased significantly by using ZnO-NPs [8]. ZnO-NPs enhanced the germination and growth of different crops [9]. ZnO-NPs positively augmented the growth and yield parameters along with pod number in *Arachis hypogaea* [10]. The rate of germination in peanut plant also increased through the application of ZnO-NPs [11]. The exogenous application of ZnO-NPs was conducive to penetration into seeds and enhanced water uptake, thus promoting seed germination [12]. The deficiency of Zn probably enhanced the level of reactive oxygen species (ROS), while the application of ZnO-NPs could relieve the damage caused by ROS because it increased the activity of antioxidative enzymes [13].

Tea (*Camellia sinensis*) is widely consumed around the world for its good taste and health-promoting properties [14,15]. The chemical composition of tea mainly consists of alkaloids, amino acids, polysaccharides, vitamins, volatile compounds, mineral elements, and polyphenols, all of which have unique sensory properties and healthy features [14]. For example, one of the main components of tea polyphenols is flavonoid, and in addition, tea polyphenols have been associated with anti-cancer, anti-microbial, anti-inflammatory, and anti-atherosclerosis properties. A major part of the health benefits of tea is attributed to tea polyphenols [16]. Total polyphenols and amino acids affect the bitterness and astringency of the green tea, while total soluble sugar could improve the sweetness and mellow taste [17]. As green tea is produced without strong oxidation, its quality depends more on the inherent chemical composition of the fresh leaves [18]. Tea quality is a cumulative property that manifests itself as a combination of a series of metabolites. Meanwhile, tea quality is also affected by the production process. Considering the various advantages of tea, it is helpful to study the changes in physiological indexes and regulation measures during its growth and development.

As a foliar fertilizer, ZnO-NPs have been studied on other plants, but there are few studies on the effect of ZnO-NPs on the physiological characteristics of tea. Thus, we adopted different concentrations of ZnO-NPs to spray tea leaves to fill this gap. The objectives of this study are to (1) investigate how ZnO-NPs influence tea growth by modulating antioxidant activity, (2) explore the influence of ZnO-NPs on secondary metabolites of the tea leaves, and (3) examine the interacting effect of ZnO-NPs on aromatic components in the tea leaves.

## 2. Materials and Methods

### 2.1. Overview of the Experimental Site and Plant Materials

The plant material used in our experiment was one variety of *C. sinensis* 'Wu niu zao', which was a small-leaf tea tree. The sampling site of this experiment is located in the Tea Garden, Xia Shu Forest Farm, Zhenjiang City, Jiangsu Province (119°13' E, 32°7' N). It belongs to the north subtropical monsoon climate zone, with sufficient light, abundant water and heat resources, and has good conditions for the development of forestry production.

### 2.2. Experiment Design

ZnO-NPs (99.8% metal basis, $90 \pm 10$ nm) purchased from Shanghai Macklin Biochemical Technology Co., Ltd. (Shanghai, China), were used in this study. ZnO-NPs were dissolved in a small volume of alcohol and then diluted with distilled water into different concentrations. Six treatments, including $0$ mg·L$^{-1}$ (CK), $10$ mg·L$^{-1}$, $50$ mg·L$^{-1}$, $100$ mg·L$^{-1}$, $150$ mg·L$^{-1}$ and $200$ mg·L$^{-1}$ of ZnO-NPs, were designed, and a total of six plots were created for each treatment containing $5$ m$^2$ of tea trees each. The experiment was started on 24–26 March, 2021, with 1 L of ZnO-NPs sprayed on per treatment. The experiment was set up with three replications.

Previous studies have suggested that high concentrations of ZnO-NPs may be toxic [19,20]. However, in our unpublished data, the Zn contents in 0.1 g of dry tea leaves under all treatments were well below the normal Zn content in tea leaves. Hence, the concentrations of ZnO-NPs selected in this experiment would not bring about heavy metal contamination of tea trees, and the tea leaves could meet the drinking standard for making tea drinks.

### 2.3. Sampling and Analysis

Sampling was performed on 1 April and 8 April. One bud and two young leaves were picked as the test material, and the fresh leaves were packed into ice boxes (refrigerated at 4 °C) and immediately transported to laboratory. The leaves for determining aroma components were firstly examined, and the remaining leaves were stored in an ultra-low temperature refrigerator (−80 °C) for a later analysis of biochemical components (including malondialdehyde (MDA), peroxidase (POD), superoxide dismutase (SOD), chlorophyll, soluble sugar, soluble protein, and tea polyphenols).

### 2.4. Determination of Main Biochemical Components

Chlorophyll: 10 tea tree leaves were selected and cut into pieces (near the main vein), and 0.2 g of pieces were put in a triangular flask with 25 mL of 96% ethanol, sealing and protecting from light for 24 h. After the pieces turned white, 2 mL of the supernatant and 4 mL of 96% ethanol were mixed, then 96% ethanol was taken as the blank, and the absorbance at wavelengths of 665 nm and 649 nm was measured. Chlorophyll a, chlorophyll b and total chlorophyll were calculated according to Li [21].

Soluble sugar: 0.2 g of tea leaves sample in each replicate were ground and diluted to 10 mL. After extracting in boiling water for 30 min, filtering the extract into a 25 mL volumetric flask, repeated this step twice, then diluted to the mark with distilled water. A total of 0.2 mL of the extraction, 1.8 mL of distilled water, 0.5 mL of anthrone ethyl acetate, and 5 mL of 98% $H_2SO_4$ were added and fully mixed using a shaker. Then measured the OD value at a wavelength of 630 nm by a Beckman DU 800 UV–visible spectrophotometer (Beckman Coulter, Inc., Brea, CA, USA, the same hereafter). The soluble sugar content was calculated according to the method described by Wu et al. [22].

Soluble protein: 0.2 g of tea leaves were weighed in each sample, then grinding it and diluting to 5 mL. Extraction was obtained after centrifuging at 8000 r/min for 15 min at 4 °C (Allegra X-22R, F1010 Rotor, Beckman Coulter, Inc., Brea, CA, USA). A total of 1 mL of extraction was taken, and 5 mL of Coomassie brilliant blue G-250 was added, mixing them thoroughly, the soluble protein content was measured in a spectrophotometer at a wavelength of 560 nm, which was according to Bradford [23].

Tea polyphenols: 0.2 g dried and crushed tea leave samples were dissolved in 5 mL of 70% methanol. After boiling and centrifuging at 3600 r/min for 10 min twice, the supernatant was gained. A total of 1 mL of gallic acid working solution, 2 mL of distilled water, and 1 mL of test solution were added into the pipette, and then 5 mL of the prepared Folin's phenol (10 mL of distilled water) were added to the pipette, after reacting 3–8 min, then 4 mL $Na_2CO_3$ were diluted to 100 mL. Tea polyphenols were estimated with a spectrophotometer at 765 nm. The content of tea polyphenols was calculated according to Li [21].

POD, SOD, and MDA: 0.2 g tea leaves were ground with 5 mL of phosphate buffer (pH = 7.8) to obtain a homogenate. POD activity was determined by guaiacol method [24]: 0.1 mL of the supernatant and 2.9 mL of the reaction solution (containing 28 μL of guaiacol and 19 μL of hydrogen peroxide) were mixed and estimated with spectrophotometer at 470 nm. SOD activity was determined by nitro-tetrazolium-blue-chloride (NBT) photo-chemical reduction method [25], firstly, transferring the homogenate to a centrifuge tube and centrifuging it at 10,000 rpm for 20 min and then configuring the reaction system, which was made up by 1.5 mL of 50 mmol/L phosphate buffer, 0.3 mL of 130 mmol/L methionine, 0.3 mL of 750 μmol/L nitro blue tetrazolium, 0.3 mL of 100 μmol/L ethylene diamine tetraacetic acid, 0.3 mL of 20 μmol/L riboflavin, and 50 μL of phosphate buffer or supernatant. Finally, we estimated SOD activity using a spectrophotometer at 560 nm. MDA content was determined by barbituric acid colorimetry [26]. Firstly, we centrifuged the homogenate at 3000 rpm for 20 min, then added 2 mL of supernatant and 2 mL of 0.6% thiobarbituric acid to the tube and put it into boiling water for 20 min. Finally, the absorbance of the supernatant was measured at 450 nm, 532 nm, and 600 nm.

*2.5. Determination of Aroma Components in Fresh Tea Leaves Using Headspace Solid-Phase Microextraction Coupled with Gas Chromatography-Mass Spectrometry (HS-SPME/GC-MS)*

Extraction of aroma components: 2 g of fresh tea leaves in each sample was chopped into pieces and put in 20 mL headspace vials. Then under the constant temperature of 50 °C, these headspace vials were shaken and equilibrated for 30 min. After that, the 65 μM PDMS-DVB (Supelco, Bellefonte, PA, USA) extraction fiber was inserted into the sample headspace vial. Headspace extracting lasted for 34 min. Then the substance was decomposed at 220 °C for 3 min, using GC-MS for separation and identification. The extraction heads were heated before and after extraction.

The GC conditions in our experiment were as follows: a GC system (Thermo Fisher Scientific Inc., Waltham, MA, USA) was equipped with a DB-5MS fused silica capillary column (30 m, 0.25 mm, 0.25 μM, Agilent Technologies, Santa Clara, CA, USA). Helium was used as carrier gas with a flow rate of 1 mL/min. The oven temperature was initially 50 °C for 2 min and then was gradually increased at 5 °C per min to 200 °C with no hold, after that the temperature was increased from 10 °C per min to 220 °C.

The MS conditions in our experiment were as follows: The electron ionization mode of the MS was operated at 70 eV, generating a scan range of 45–450 amu. The ion source and transfer line were at 200 °C and 250 °C.

The spectra and retention times of peaks were recorded, and the aroma compounds were identified using X-caliber with reference to the National Institute of Standards and Technology (NIST)'s 12th library and chemistry database. The relative content of these components was calculated by peak area normalization measurements [27]:

$$\text{Aroma components concentration (\%)} = \frac{M}{N} \times 100\% \tag{1}$$

M, the peak area of the individual compounds; N, the total peak area.

*2.6. Data Collection and Analysis*

The values were expressed as mean ± SD for three replicates. All statistics were processed by Excel (Office 2019 Pro Plus, Microsoft Corporation, Redmond, WA, USA), and one-way analysis of variance (ANOVA) was performed using SPSS 26.0 (IBM, Armonk, NY, USA) followed by Duncan's multiple range test. The *p*-values less than 0.05 were considered to indicate significance within groups..

**3. Results**

*3.1. Chlorophyll Content under Different Concentrations of ZnO-NPs*

As shown in Figure 1A,B, foliar spraying of different concentrations of ZnO-NPs had no significant effect on the chlorophyll content in the fresh tea leaves. In the samples picked on 1 April, only the 10 mg·L$^{-1}$ and 150 mg·L$^{-1}$ concentrations of ZnO-NPs had a higher chlorophyll a and chlorophyll b content, which was higher than that of CK, but the difference was not significant. In the samples picked on 8 April, the chlorophyll a and chlorophyll b contents of all fresh tea leaves treated with ZnO-NPs were lower than that of CK. The changes in the content of chlorophyll a and chlorophyll b displayed the same trend as the changes in the total amount of chlorophyll.

Different treatments had different effects on total chlorophyll content in the fresh tea leaves (Figure 1C). Seven days after spraying ZnO-NPs fertilizer, the total chlorophyll content in the fresh tea leaves treated with 150 mg·L$^{-1}$ and 10 mg·L$^{-1}$ was higher than that of CK; however, the difference was not significant among all treatments. Under 150 mg·L$^{-1}$ treatment, the content of total chlorophyll in the fresh leaves was the highest (6.84 mg·g$^{-1}$), and the content of chlorophyll b reached the maximum of 1.46 mg·g$^{-1}$ under the 10 mg·L$^{-1}$ concentration. Fourteen days after spraying the ZnO-NPs fertilizer, the total chlorophyll content of the fresh tea leaves treated with ZnO-NPs was obviously lower than that of the CK.

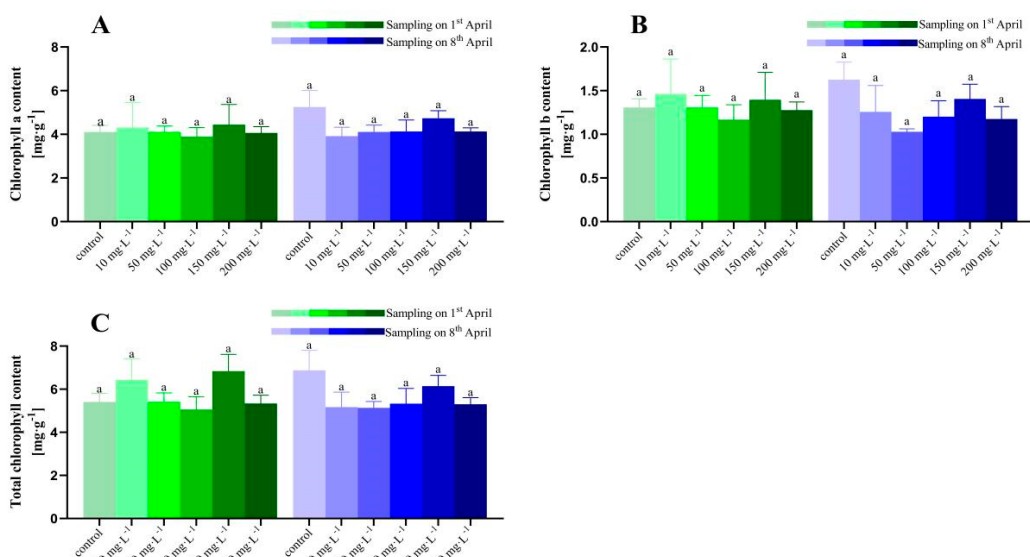

**Figure 1.** Effect of zinc oxide nanoparticles on chlorophyll a (**A**), chlorophyll b (**B**) and total chlorophyll (**C**) content of tea leaves. Different lowercase letters indicated significant differences at the 0.05 level.

### 3.2. Tea Nutritional Quality under Different Concentrations of ZnO-NPs

The application of ZnO-NPs caused significant effects on soluble sugar, soluble protein, and tea polyphenol contents. As a fast-acting nutrient, they have more beneficial to the vegetative growth of tea trees in a short time.

As can be seen in Figure 2A, there were significant differences in soluble sugar content on 1 April. Soluble sugar content under the 10 mg·L$^{-1}$ ZnO-NPs treatment (47.89 mg·g$^{-1}$) was significantly higher than that in all other ZnO-NPs treatments, but had no significant difference with that of CK. Under 50, 150, 200 mg·L$^{-1}$, the soluble sugar content was close with no significant difference on 1 April. The difference of soluble sugar content in the fresh tea leaves increased with the increasing of the ZnO-NPs concentrations. The 50 mg·L$^{-1}$ treatment caused the lowest content in two sampling periods. Samples which were picked on 8 April had the similar trend, but the contents at 0, 50, and 100 mg·L$^{-1}$ were all lower than those of the samples which were picked on 1 April.

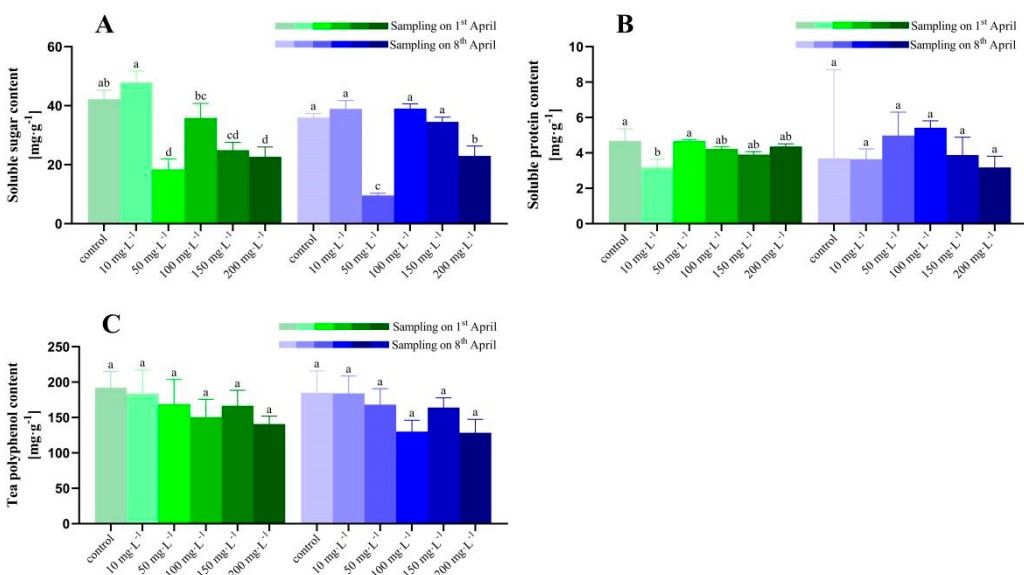

**Figure 2.** Effect of zinc oxide nanoparticles on soluble sugar (**A**), soluble protein (**B**) and polyphenol (**C**) content of tea leaves. Different lowercase letters indicated significant differences at the 0.05 level.

Significant differences were found in soluble protein content in the fresh tea leaves which were picked on 1 April. The soluble protein content in the fresh tea leaves at 50 mg·L$^{-1}$ and in the CK was higher in the first sampling period (4.66 mg·g$^{-1}$ and 4.67 mg·g$^{-1}$, respectively), which was significantly higher than that under the 10 mg·L$^{-1}$ treatment. The soluble protein content in the fresh tea leaves which were collected on 8 April did not have significant differences with that of the other treatments. In this period, the soluble protein content in the CK was 3.68 mg·g$^{-1}$, and the contents under the 50 mg·L$^{-1}$ and 100 mg·L$^{-1}$ treatments were much higher than those in the CK.

Figure 2C shows the effects of different ZnO-NPs treatments on the content of tea polyphenols in the tea leaves. In general, the ZnO-NPs use brought about a limited effect on the content of tea polyphenols, and there was no significant difference among the treatments in both periods. Among the tea samples picked in the two periods, the CK tea samples had the highest content of tea polyphenols, which was 192.13 mg·g$^{-1}$ and 184.81 mg·g$^{-1}$, respectively. Then the content of tea polyphenols decreased generally with the increasing of ZnO-NPs concentrations. The content of tea polyphenols in the tea leaves treated with 100 mg·L$^{-1}$ was lower in the two periods, while the content of tea polyphenols in leaves treated with 10 mg·L$^{-1}$ and in the CK was the opposite.

### 3.3. Biochemical Components of Tea Leaves under Different Concentrations of ZnO-NPs

The experimental results demonstrated that under different concentrations of ZnO-NPs, the MDA content, POD and SOD activity (Figure 3) in the fresh tea leaves (picked on 1 April) were significantly different. The differences in MDA content and SOD activity in the fresh tea leaves picked on 8 April were highly significant, while the differences in POD activity were not significant. The experimental results showed that the foliar spraying of the ZnO-NPs solution with a concentration of 100 mg·L$^{-1}$ was beneficial to prevent membrane lipid peroxidation of the tea trees.

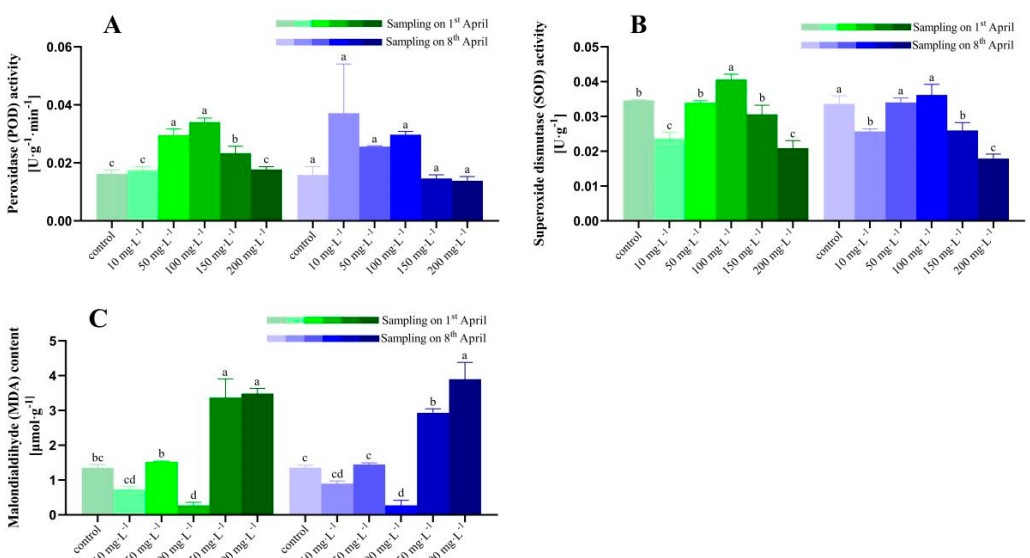

**Figure 3.** Effects of zinc oxide nanoparticles on antioxidant enzyme activities (**A**,**B**) and MDA content (**C**) in tea leaves. Different lowercase letters indicated significant differences at the 0.05 level.

Figure 3A reflects the effect of different ZnO-NPs treatments on the POD activity in fresh leaves of tea plants. Seven days after spraying, POD activity in the fresh leaves differed significantly from one treatment to another, with an 'A' trend. POD activity was the lowest at 0.0162 U·g$^{-1}$·min$^{-1}$ in CK, and then increased with the increasing of the ZnO-NPs concentration, reaching the maximum at 100 mg·L$^{-1}$ (0.034 U·g$^{-1}$·min$^{-1}$), which was significantly higher than in the other treatments. After that, POD activity started to decrease and reached the lowest at 200 mg·L$^{-1}$ (0.0177 U·g$^{-1}$·min$^{-1}$), but it was still higher than that of CK. Fourteen days after spraying, the difference in POD activity in the fresh tea

leaves was not significant, but decreased in a fluctuating manner with the increasing of the ZnO-NPs concentration. POD activity was greater at 10 mg·L$^{-1}$ and 100 mg·L$^{-1}$, which was 0.0371 U·g$^{-1}$·min$^{-1}$ and 0.0287 U·g$^{-1}$·min$^{-1}$, respectively. POD activity in the fresh leaves treated with 150 mg·L$^{-1}$ and 200 mg·L$^{-1}$ of ZnO-NPs was lower than that of CK.

Figure 3B describes the impact of different exogenous treatments on SOD activity in the fresh tea leaves. SOD activity in the fresh tea leaves of the CK treatment was higher (0.0346 U·g$^{-1}$·min$^{-1}$); while it fluctuated with the increase in ZnO-NPs concentration and reached a maximum of 0.0406 U·g$^{-1}$·min$^{-1}$ at 100 mg·L$^{-1}$. The results of SOD activity in tea leaves at fourteen days after spraying were similar to those at seven days after spraying, and the highest SOD activity in the fresh tea leaves was also found under 100 mg·L$^{-1}$ (0.0362 U·g$^{-1}$·min$^{-1}$), followed by CK (0.0336 U·g$^{-1}$·min$^{-1}$). SOD activity in the rest of the treatments was lower than that in the CK.

As shown in Figure 3C, the trend of MDA content in the fresh tea leaves treated with ZnO-NPs was the same in different sampling periods, and the content varied significantly among these treatments. In the tea samples picked seven days after spraying, MDA content in leaves at 100 mg·L$^{-1}$ was the lowest at 0.267 μmol·g$^{-1}$, which was significantly lower than that in the other treatments. The MDA contents in the fresh tea leaves at 150 mg·L$^{-1}$ and 200 mg·L$^{-1}$ concentrations were higher, 3.369 μmol·g$^{-1}$ and 3.484 μmol·g$^{-1}$, respectively. On 8 April, MDA content in the tea leaves from the CK was 1.355 μmol·g$^{-1}$, which was significantly higher than that of the leaves treated with 100 mg·L$^{-1}$ ZnO-NPs, but significantly lower than that of the leaves treated with 150 mg·L$^{-1}$ and 200 mg·L$^{-1}$. MDA content dropped to a minimum value of 0.2724 μmol·g$^{-1}$ at 100 mg·L$^{-1}$. Among the six treatments, 150 mg·L$^{-1}$ and 200 mg·L$^{-1}$ of ZnO-NPs had a greater effect on MDA content in the fresh leaves of tea trees.

### 3.4. Aroma Components of Tea Leaves under Different Concentrations of ZnO-NPs

Statistics illustrated that a total of twenty-seven compounds were detected in fresh leaves (Table 1). In comparison to the CK, twelve compounds were detected at the concentration of 10 mg·L$^{-1}$, and the most abundant compound was benzene, followed by bestatin. The 50 mg·L$^{-1}$ ZnO-NPs treatment induced the greatest number compounds, which was sixteen, and the main compound was benzene, followed by trans-3-Hexenyl acetate and bestatin, while sarcosine and γ-Terpinene were unique compounds. Fourteen compounds were identified at 100 mg·L$^{-1}$, the major compounds were benzene, ketorolac, ocimene and linalool, and ocimene as well as farnesene were specific compounds. Eleven compounds were detected at 150 mg·L$^{-1}$, with benzene being the highest, followed by linalool and leaf alcohol. Only one compound, benzene, was detected at 200 mg·L$^{-1}$. Benzene was the common aroma compound in the fresh tea leaves among all treatments.

**Table 1.** Aroma compounds and relative contents in fresh tea leaves under different treatments.

| No. | Compounds | RI | Relative Content (%) Treatments | | | | | |
|---|---|---|---|---|---|---|---|---|
| | | | CK | 10 mg·L$^{-1}$ | 50 mg·L$^{-1}$ | 100 mg·L$^{-1}$ | 150 mg·L$^{-1}$ | 200 mg·L$^{-1}$ |
| 1 | Ethanolamine | 698.92 | 6.53 ± 0.36 a | 4.47 ± 0.50 a | 3.68 ± 1.28 b | - | 5.65 ± 0.26 a | - |
| 2 | H-Ala-Beta-Ala-Oh | 805.50 | - | 4.81 ± 0.66 a | 1.96 ± 0.48 b | - | - | - |
| 3 | Sarcosine | 839.45 | - | - | 1.02 ± 0.65 | - | - | - |
| 4 | Benzene | 953.66 | 54.78 ± 13.23 b | 49.93 ± 8.45 c | 49.4 ± 6.59 c | 30.84 ± 4.98 d | 62.14 ± 14.21 a | 28.72 ± 5.32 d |
| 5 | Bestatin | 1030.41 | - | 17.78 ± 3.45 a | 5.98 ± 2.37 b | - | 3.28 ± 1.07 c | - |
| 6 | Ketorolac | 1031.09 | 11.83 ± 0.69 | - | - | 11.8 ± 1.02 | - | - |
| 7 | γ-Terpinene | 1058.32 | - | - | 0.52 ± 0.18 | - | - | - |
| 8 | Ocimene | 1259.86 | - | - | - | 8.92 ± 4.12 | - | - |
| 9 | Ocimene mixture of isomers | 1276.41 | - | - | 0.51 ± 0.26 | - | 0.6 ± 0.23 | - |
| 10 | Esculetin | 1319.01 | 1.04 ± 0.57 a | 1.8 ± 1.23 a | 0.63 ± 0.21 b | - | - | - |
| 11 | Anisole | 1325.15 | - | - | - | 2.66 ± 1.21 | 2.09 ± 1.02 | - |
| 12 | *trans*-3-hexen-1-ol | 1331.08 | - | 3.64 ± 2.26 a | 1.57 ± 0.75 b | 1.3 ± 0.52 b | - | - |
| 13 | *cis*-3-hexenyl acetate | 1336.21 | 8.98 ± 3.12 a | 3.78 ± 1.21 c | - | 4.95 ± 2.21 b | 6.07 ± 2.25 b | - |
| 14 | *trans*-3-hexenyl acetate | 1351.08 | - | - | 12.33 ± 6.24 a | 4.78 ± 0.85 b | - | - |
| 15 | Leaf alcohol | 1351.25 | 1.8 ± 1.25 b | - | 1.52 ± 0.64 b | - | 6.53 ± 2.25 a | - |
| 16 | *trans*-2-hexenyl acetate | 1354.77 | - | - | 0.42 ± 0.14 | - | 0.69 ± 0.29 | - |
| 17 | 3-(Chloromethyl)heptane | 1476.84 | - | 0.88 ± 0.24 b | - | 1.27 ± 1.07 a | - | - |
| 18 | Fema 3498 | 1494.38 | 0.93 ± 0.35 b | - | 1.03 ± 0.45 b | 3.26 ± 1.22 a | - | - |
| 19 | *cis*-alpha,alpha,5-trimethyl-5-vinyltetrahydrofuran-2-methanol | 1524.89 | 1.51 ± 0.96 b | 3.69 ± 1.82 a | - | - | 0.78 ± 0.26 c | - |
| 20 | Linalool | 1545.78 | 5.97 ± 1.24 b | 4.09 ± 1.08 c | - | 5.77 ± 1.35 b | 6.75 ± 2.21 a | - |
| 21 | *cis*-3-Hexenyl 2-methylbutanoate | 1657.69 | - | - | 1.08 ± 0.37 b | 3.45 ± 1.12 a | 0.56 ± 0.17 c | - |
| 22 | methyl (Z)-3,7-dimethylocta-2,6-dienoate | 1657.75 | 0.77 ± 0.24 b | - | - | 1.02 ± 0.65 a | - | - |
| 23 | Methyl salicylate | 1662.75 | 0.97 ± 0.47 | - | - | - | - | - |
| 24 | *cis*-3-Hexenyl hexanoate | 1665.66 | 1.54 ± 1.21 a | 0.94 ± 0.27 b | 1.7 ± 0.28 a | - | - | - |
| 25 | *cis*-3-Hexenyl butyrate | 1665.69 | 1.11 ± 0.56 b | - | 1.37 ± 0.49 b | 2.92 ± 0.98 a | - | - |
| 26 | Farnesene | 1768.47 | - | - | - | 3.94 ± 0.74 | - | - |
| 27 | 1-Hexadecanol | 1839.39 | 0.89 ± 0.64 b | 1.52 ± 0.89 a | - | - | - | - |

Note: "-" indicated that scent compounds were not detected. All data in the table were average of three replicates ± SD; the letters in the table indicated significant differences in Duncan's multiple comparisons ($p < 0.05$); RI indicated that retention indices of volatiles were calculated using an alkanes standard (C8–C30).

## 4. Discussion

ZnO-NPs have positive effects on plants as they could alter the physical, chemical, and biological characteristics, as well as the catalytic properties of plants, thereby affecting their physiological and biochemical activities. In our study, ZnO-NPs sprayed on tea trees improved the chlorophyll, soluble sugar, soluble protein, tea polyphenol, and MDA contents and POD and SOD activities (Figures 1–3).

Chlorophyll is the primary site of photosynthesis in green plants, and its content is a significant index reflecting the performance of plant photosynthesis. Rajiya et al. [10] pointed out that exogenous ZnO-NPs increased the chlorophyll content of guar (*Cyamopsis tetragonoloba* L.). Faizan and Hayat [28] found that chlorophyll content of tomato (*Lycopersicon esculentum*) was significantly increased after spraying various concentrations of ZnO-NPs on the leaves, which was similar to our result. Ahmed et al. [29] expounded that the application of ZnO-NPs promoted the nutritional growth of tomato seedlings, and their total chlorophyll content increased with the increase in the concentration of ZnO-NPs. In this study, ZnO-NPs increased the content of chlorophyll in a concentration-dependent manner, with plants treated with 150 mg·L$^{-1}$ having the highest chlorophyll content and samples treated with other concentrations (10, 50, 100, or 200 mg·L$^{-1}$) having lower levels of chlorophyll content. The chlorophyll content increased due to the fact that ZnO-NPs may induce the translation and/or transcription of the enzyme genes involved in the biosynthesis of chlorophyll. Similar increases in chlorophyll content by the application of other NPs have also been documented [30–32]. The plant photosynthetic capacity could be represented by the chlorophyll content. This study revealed that the foliar spray of ZnO-NPs boosted the accumulation of chlorophyll to its maximum in the tea tree leaves. The chlorophyll content measured in the fresh leaves was linearly correlated with the ZnO-NPs concentration, and the chlorophyll content increased generally with the increasing ZnO-NPs concentrations, but chlorophyll synthesis was inhibited when the ZnO-NPs concentration was too high. Chlorophyll a and total chlorophyll in the fresh leaves of tea trees reached the maximum when the concentration of ZnO-NPs was 150 mg·L$^{-1}$, with a lower chlorophyll content under other treatments (10, 50, 100, and 200 mg·L$^{-1}$), and the chlorophyll b content did not change significantly. The possible reason is that ZnO-NPs can enter the chlorophyll biosynthesis process, resulting in an increase in the rate of translation or transcription of chlorophyll-related genes and an increased pigment content in the leaves [33]. It may also be explained by the presence of ZnO-NPs affecting the activity of related enzymes in the chlorophyll biosynthesis process or decoding the related proteins affecting chlorophyll synthesis, which accelerates the rate of chlorophyll synthesis and increases the chlorophyll content in the leaves.

Soluble protein is a crucial osmoregulatory substance in plant leaves and one of the most essential energy sources for plant growth and development [26]. In this study, it was observed that foliar spraying of ZnO-NPs on the tea trees could increase the soluble protein content in the fresh leaves. This may be because ZnO-NPs can affect the structure or protein activity via binding proteins and sterols directly, thus regulating the activity of proteins and other enzymes in the membrane, maintaining protein structure and regulating the enzymes responsible for protein synthesis [34]. The same conclusion was reached by Raliya and Tarafdar [35] and Mukherjee et al. [36].

Soluble sugar is a vital substance for plants to maintain the concentration of cellular fluid in the body and improve water retention. They are also factors that affect the flavor of tea leaves. Tea polyphenols are the main source of bitter taste in tea, and their content determines the freshness of tea broth [37]. Studies have shown that external measures induce tea tree growth, during which tea tree genes may change, and new mRNAs and proteins are synthesized, affecting the relevant metabolic processes [38]. In this study, the soluble sugar and tea polyphenol contents in the tea leaves did not change significantly with the increasing concentration of ZnO-NPs, but were essentially lower than those in the CK. This may be because the presence of ZnO-NPs affected the expression of genes related to sugar metabolism and tea polyphenol synthesis in tea trees, which reduced the rate of

soluble sugar metabolism as well as tea polyphenol synthesis, resulting in low soluble sugar content and tea polyphenol content in the fresh leaves.

Higher level of ROS is caused by abnormal cellular metabolism, which leads to oxidative injury to macromolecules and ultimately leads to cell death [39]. The accumulation of ROS in plants indicates that lipids and proteins are seriously damaged, and the normal metabolic process in plants is destroyed, thus resulting in a decline in the plant biomass [40]. When subjected to such stress, plants activate non-enzymatic and enzymatic antioxidant systems to reduce the negative effect caused by the stress [41]. A previous study confirmed that exogenous ZnO-NPs elevated the enzymatic defense mechanisms by decreasing the MDA content, and increasing POD and SOD activities [28]. Furthermore, ZnO-NPs significantly enhanced the activity of antioxidant enzymes (POD and SOD) in the current study, especially in the case of ZnO-NPs at 100 mg·L$^{-1}$. It is well documented that Zn plays a critical role in balancing the stability of bio-membranes and proteins by balancing the scavenging ROS production [42]. These evidences confirmed our experimental results that the application of ZnO-NPs enhanced the enzymatic antioxidant system (POD and SOD) to scavenge the excessive ROS in tea trees.

Aroma components are secondary metabolites that are composed of low molecular volatiles. These components can be divided into volatile terpenes (VTs), volatile fatty-acid decompositions (VFADs), and volatile phenylpropanoid/benzenoid (VPBs) [43]. The aroma components in the tea leaves account for a small proportion of the dry matter of tea, but have a crucial impact on the tea quality. Differences in the types and contents of aroma components in fresh tea leaves are the basis for the formation of different tea aroma types [44]. How heavy metals affect the growth of tea trees is also well researched. A large number of studies have shown that metal elements such as copper and zinc have physiological and biochemical effects on tea trees. The application of zinc can increase the content of selenium, caffeine, and amino acids in teas and boost tea quality [45]. However, little research has been conducted on the effect of zinc fertilizer on aroma components in fresh tea leaves. Cai [46] indicated that the application of zinc fertilizer could affect the content of benzyl alcohol and benzyl alcohol primrose glycosides in fresh tea leaves. One of the possible mechanisms by which tea plants respond to zinc fertilizer is the regulation of glycosyltransferase gene expression to increase the aroma glycosides in tea plants. It has been reported that the saturated and unsaturated aldehydes as well as alcohols aroma compounds are the primary sources of fresh green aroma in tea [47]. Fu et al. [48] proposed that external treatment enhanced the content of terpenes and fatty-acid-derived aroma substances in fresh leaves of tea plants. In this study, when the concentration of ZnO-NPs was relatively high, fewer types of aroma substances were present in the fresh tea leaves, and only one aromatic compound was measured at 200 mg·L$^{-1}$ of ZnO-NPs. The total number of aroma components measured in other treatments was similar, being they were mainly VTs and VFADs. The content of terpene aroma compounds basically increased with the increase in the concentration of ZnO-NPs, which may be because the higher concentration of zinc stimulated the upregulation of terpenoid-synthesis-related gene expression and the hydrolysis of terpene glycosides. Terpenes are known to have great potential in plant defense and strong antioxidant activity. Until now, they have been widely utilized in producing medicine and food additives [49,50]. The contents of fatty-acid-derived aroma compounds did not change significantly with the increasing concentration of ZnO-NPs, but the types and contents of fatty-acid-derived aroma components were higher at the two intermediate concentrations of 100 mg·L$^{-1}$ and 150 mg·L$^{-1}$. The formation mechanism of aroma components from various fatty acid sources may be diverse, and the effects of ZnO-NPs on different volatiles are also discrepant.

## 5. Conclusions

The ZnO-NPs application is a novel and potential way to improve plant growth and development. Through this study, we concluded that ZnO-NPs-mediated response was concentration-dependent. In general, tea leaves treated with 100 mg·L$^{-1}$ of ZnO-NPs

exhibited the most promising response and increased the activities of POD and SOD, as well as the contents of soluble sugar, soluble proteins, chlorophyll, and tea polyphenols, while other concentrations of ZnO-NPs varied significantly. Antioxidant systems and soluble protein content enhanced by 100 mg·L$^{-1}$ of ZnO-NPs provided better ROS protection to plants. Hence, the ZnO-NPs at the 100 mg·L$^{-1}$ concentration for the leaf treatment could be considered an efficient nano-micronutrient for tea trees to boost the yield and improve the tea quality. With the deepening of the research on the mechanism of ZnO-NPs regulation of plant growth and development, some high-throughput technologies, such as genomics, transcriptomics, and proteomics, have been applied to isolate and identify the related genes, so as to clarify the regulatory mechanism of ZnO-NPs and achieve precise regulation of plant growth and development.

**Author Contributions:** Conceptualization, C.C. and J.L.; methodology, F.Y.; software, J.L.; formal analysis, C.C.; investigation, C.C., J.L. and H.C.; writing-original draft preparation, C.C. and J.L.; writing-review and editing, F.Y.; funding acquisition, C.C. All authors have read and agreed to the published version of the manuscript.

**Funding:** This research was funded by Yangzhou 'Lvyangjinfeng' excellent doctoral project (YZ-LYJFJH2022YXBS019).

**Acknowledgments:** We would like to thank Wang Jihua and Nanjing Yangzi Jasmine Valley Culture Technology Co., Ltd. for providing the test materials.

**Conflicts of Interest:** The authors declare no conflict of interest.

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
