# Peer review of "Zinc Oxide Nanoparticles Enhanced Growth of Tea Trees via Modulating Antioxidant Activity and Secondary Metabolites"

_horticulturae, doi:10.3390/horticulturae9060631_

Round 1
Reviewer 1 Report
Manuscript ID: horticulturae-2367956
Type: Article
Title: Zinc Oxide Nanoparticles Enhanced Growth of Tea Trees via Modulating Antioxidant Activity and Secondary Metabolites
Recommendation: Minor Revision
Excess use of chemical fertilizers and pesticides has shown negative impacts on soil health and environment. Chemical residues destroy the soil ecosystem and crop productivity by affecting the non-targeted organisms and decrease soil fertility simultaneously. To prevent such adversity in soil and environment, researchers have applied nanotechnology in agriculture. Nanotechnology is the rising technology of the current decade, which has shown promising results in controlling excess agri-inputs and maintaining environmental balance. Zinc oxide nanoparticles (ZnO-NPs) have excellent biological properties and are expected to be an ideal choice for plant zinc fertilizer. Acknowledging the importance of NPs as “Nano-fertilizer”, the current study was conducted to assess the effect of ZnO-NPs on tea plant by spraying them on leaves. The volatile substances, aroma, biochemical properties and antioxidant behaviour of ZnO-NPs-stressed tea leaves were assessed. The authors have observed that varying concentrations of ZnO-NPs on tea plants differed variably. It was found that ZnO-NPs at 150 ppm caused the increase in chlorophyll pigment, while 100 ppm promoted the accumulation of soluble protein and the activity of antioxidant enzymes which however, reduced the accumulation of MDA. Application of NPs showed a considerable reduction in tea polyphenols content. Overall, the manuscript is good and very suitable for the journal horticulture.
1. The title is eye-catching and interesting.
2. This article is written very well and may helpful for readers around the globe.
3. Abstract is good and to the point.
4. Introduction section is written very well.
5. The material and methods section has described in detailed. From where ZnO-NPs obtained? Are they chemically/biologically synthesized? Or procured commercially?
6. Results and discussion are appropriately discussed. The overall manuscript is good. However, there are some errors have been observed which is needed to improve before publication of the article.
7. Tables and figures are nicely presented. Table 1 showing Aroma compounds and relative contents in fresh tea leaves under different treatments is very good.
8. Conclusion section is good. However, I think it needs revision/modifications. It seems like the authors have pasted the contents from the abstracts section. I am sorry so for these comments. But the conclusion section needs refining before the final publication of this article.

Reviewer 2 Report
From reviewing your work “Zinc Oxide Nanoparticles Enhanced Growth of Tea Trees via Modulating Antioxidant Activity and Secondary Metabolites”, I found it interesting and the results are technically sound, however some points should be improved, and others clarified so that the work can be accepted in this famous and prestigious journal.
1) The language of the manuscript needs to be revised.
2) The abbreviation (MDA) should be fully named “malondialdehyde” in the abstract.
3) Line 94, “Previous research demonstrated that high concentration of ZnO-NPs may be toxic.”, missed some supported references.
4) In section “2-Materials and Methods”. Brief descriptions of the state-of-art of ZnO-NPs concerning its purchase or preparation technique, and how much the nano-size of ZnO particles, have to be included.
5) The authors stated in section “2.4. Determination of main biochemical components” the absorption values of Chlorophyll, OD, Tea polyphenols, POD, SOD and MDA. It is better to show the absorption curves of these measurements.
6) In 2.5. “Determination of aroma components in fresh tea leaves using HS-SPME/GC-MS”. The full name of (HS-SPME) “Headspace solid-phase microextraction” and (GC-MS) “Coupled to gas chromatography–mass spectrometry” should be stated.
7) Almost all figures in low resolution must be changed.
8) The discussion part from line 325 to 331 needs more evidential support references.
Moderate editing of english language and the language of the manuscript needs to be revised.Author Response
Please see the attachment

Reviewer 3 Report
All suggestions and comments are inserted into the attached file.

Minor English Language editing is required (see the attached file)
Reviewer 4 Report
In this manuscript, Chen and co-authors investigated the impact of spraying of Zinc oxide nanoparticles on volatile aroma substances and biochemical aspects of tea leaves. Experimental design is acceptable. However, results are overstated, and discussion need to further improved. There are several typographical and grammatical mistakes. Most of places, sentences are not scientifically right. Importantly, non-significant data has been linked with the positive effect of the treatment.
Following some comments authors may consider while revising the manuscript,
1. For consistency, convert the unit “ppm” to “mg. L-1”
2. In the abstract, the impact of treatment on tea volatiles is missing. Please include.
3. Abstract, line 20- Replace “but 20 decreased” with “including a decrease in”
4. Introduction, line 32-34- Rewrite the sentence, it is not clear.
5. Introduction, lines 34-39- Rewrite the sentence, it is not clear, and the sentence should be small.
6. Introduction, lines 39-41- looks irrelevant. Delete or make it relevant.
7. Introduction, line 44 - use the full form of ZnO-NP (zinc oxide nanoparticles), when it was used first time. Correct it.
8. Introduction, lines 54-55- Rewrite of sentence, it is not clear.
9. Introduction, line 56-57- I think is not making any sense. Rewrite the sentence.
10. Introduction, lines 63-64- I think this is not right “tea polyphenols are mainly metabolites of flavonoids”
11. Materials and Methods, lines 96-99- what is the mean by “not cause heavy metal contamination..”?
12. Materials and Methods, lines 103-107- Rewrite of sentence, it is not clear.
13. Impact of 150 ppm on chlorophyll content is not significant. I think no need to highlight it.
14. Please check the claims of results. If the difference is not significant, it means the outcome should not be considered as a positive.
15. Discussion section should be improved, especially in the context of volatile composition.
Need a major revision.
Round 2
Reviewer 4 Report
No further comments.
Manuscript is improved now.